

# Evaluation of alanine aminotransferase/aspartate aminotransferase ratio and high-density lipoprotein for predicting neonatal adverse outcomes associated with intrahepatic cholestasis of pregnancy

Xizhenzi Fan[1,*], Xia Li[2,*], Tianxiao Yu[1], Ruifen Jiao[3], Wenhui Song[1], Achou Su[1], Mingwei Li[1] and Qing Guo[4]

[1] Research Center for Clinical Medical Sciences, The Fourth Hospital of Shijiazhuang, Shijiazhuang, China
[2] Department of Scientific Research and Education, The Fourth Hospital of Shijiazhuang, Shijiazhuang, China
[3] Department of Obstetrics, The Fourth Hospital of Shijiazhuang, Shijiazhuang, China
[4] Department of Obstetrics, Hebei Key Laboratory of Maternal and Fetal Medicine, The Fourth Hospital of Shijiazhuang, Shijiazhuang, Shijiazhuang, China
[*] These authors contributed equally to this work.

Corresponding author
Qing Guo, gingguo123@126.com

## ABSTRACT

**Background.** To determine the association between lipid metabolism and intrahepatic cholestasis of pregnancy (ICP), and explore the value of maternal alanine aminotransferase/aspartate aminotransferase (ALT/AST) and high-density lipoprotein (HDL) in predicting adverse neonatal outcomes in women with ICP.

**Methods.** A total of 147 pregnant women with ICP admitted to The Fourth Hospital of Shijiazhuang and 120 normal pregnant women in the same period were selected in this study. The Mann-Whitney U test and Chi-square tests were used to compare the differences in clinical data. Multivariate logistic regression was used to analyze the relationship between ALT/AST and the occurrence of adverse pregnancy outcomes in patients with ICP. The combined predictive value of ALT/AST and HDL was determined by receiver operating characteristic (ROC) curve analysis.

**Results.** Among 147 women with ICP, 122 women had total bile acid (TBA) levels of 10–39.9 μmol/L, and 25 had TBA ≥ 40 μmol/L. There was significantly lower gestational age in patients with severe ICP than in those with mild and control groups (all $p < 0.05$), and the weight of newborns in the maternal ICP group was significantly lower than in the control group ($p < 0.05$). Increasing TBA levels was associated with higher AST, ALT, ALT/AST, and lower HDL level (all $p < 0.05$). Meanwhile, higher levels of ALT/AST was positively associated with neonatal hyperbilirubinemia [adjusted odds ratio (AOR) = 4.019, 95% CI [1.757–9.194, $p = 0.001$] and cardiac injury [AOR = 3.500, 95% CI [1.535–7.987], $p = 0.003$]. HDL was a significant protective factor for neonatal hyperbilirubinemia and cardiac injury [AOR = 0.315, 95% CI [0.126–0.788], $p = 0.014$; AOR = 0.134 (0.039–0.461), $p = 0.001$]. The area under the ROC curve (AUC) for prediction of neonatal hyperbilirubinemia by ALT/AST combined with HDL was 0.668 [95% CI [56.3–77.3%], $p = 0.002$], and the sensitivity and specificity were 47.1% and 84.0%, respectively. To predict neonatal cardiac injury, the AUC value was

0.668 [95% CI [56.4–77.1%], $p = 0.002$], with sensitivity and specificity were 41.2% and 87.1%, respectively.

**Conclusions**. The levels of higher ALT/AST and lower HDL were significantly associated with the risk of ICP-related adverse neonatal outcomes. Moreover, ALT/AST combined with HDL has moderate clinical value in predicting the adverse outcomes of neonatal hyperbilirubinemia and cardiac injury.

## INTRODUCTION

Intrahepatic cholestasis during pregnancy (ICP) is an idiopathic disease that occurs during pregnancy characterized by different degrees of skin pruritus and abnormal liver function (*Jurk, Kremer & Schleussner, 2021*). Although most of these symptoms subside rapidly after termination of pregnancy (*Hobson, Gandhi & Sobel, 2022*), ICP can have a significant impact on the perinatal infant because of the cytotoxic of bile acids which can interfere with fetal growth and development through the placenta, inducing adverse events such as preterm birth and fetal hypoxia (*Zhang et al., 2016*; *Devalla & Srivastava, 2022*; *Yang et al., 2022*; *Sahni & Jogdand, 2022*). Studies have shown that the risk of intrauterine mortality is significantly increased in maternal with ICP, which may be related to fetal cardiac dysfunction or arrhythmias due to elevated bile acids (*Vasavan et al., 2021*; *Zhang et al., 2023*). Unfortunately, the pathogenesis of ICP is unclear, some studies have reported that abnormalities in estrogen metabolism and lipid metabolism may be associated with the development of ICP (*Başaranoğlu et al., 2017*; *Dann et al., 2006*; *Martineau et al., 2015*).

Currently, the serum total bile acids (TBA) concentration and clinical presentation are the main indicators of ICP severity, but there are no laboratory indicators with high specificity for the diagnosis of ICP-related adverse maternal and neonatal outcomes. The treatment for ICP patients and fetuses is limited to symptomatic support and close monitoring. In addition, although ALT and AST, which are commonly used indicators to reflect the degree of liver damage, have also received widespread attention, some studies have found that serum ALT and AST are also significantly elevated in pregnant women with ICP, just as liver function impairment occurs in normal population, suggesting that serum ALT and AST can be used as indicators to evaluate ICP-related adverse pregnancy outcomes (*Juusela et al., 2020*; *Ekiz et al., 2016*; *Kushner et al., 2022*). However, there are few reports on whether the ratio of serum ALT/AST in ICP pregnant women is also higher than that in normal pregnant women and whether it is associated with ICP-related adverse maternal and neonatal outcomes. Our study aimed to provide a clinical reference for improving the neonatal prognosis in ICP mothers by comparing the clinical significance of liver function and lipid metabolism indicators in predicting ICP maternal and neonatal outcomes, and to explore the possibility of using liver enzymes and HDL to predict adverse pregnancy outcomes.

## MATERIALS & METHODS

### Study participants

This retrospective study consisted of 147 pregnant women diagnosed ICP and 120 normal pregnant women who came to The Fourth Hospital of Shijiazhuang, China, for obstetric examination between December 2016 to December 2022. All pregnant women and their families were informed in detail about the purpose of the experiment and signed informed consent at the time of enrolment. The study was approved by the Ethics Review Committee of The Fourth Hospital of Shijiazhuang (No.20230034) and all the researchers collected data in strict accordance with the ethical requirements throughout the study. Eligible participants were women aged 18-45; singleton natural pregnancy; without diabetes, hepatobiliary diseases, immunological diseases or other pregnancy complications; pregnant women were conscious and able to cooperate with the study and the data of the enrolled cases were complete. The exclusion criteria: pregnant women with combined psychiatric disorders; pregnant women with biliary obstruction and other gallbladder diseases; patients with hypothyroidism or subhypothyroidism, complicated intrauterine infections and other complications affecting the outcome of the study.

### Data collection

The sociological information and basic characteristics of all pregnant women were obtained by questionnaire, including age, height, pre-pregnancy weight, number of births, *etc*. The gestational week was determined by last menstrual period and/or ultrasonographic measurements, and the condition of the newborn was obtained by consulting delivery record and newborn care records. All blood samples were fasted for more than 8 h and collected at the same time in the morning under aseptic conditions, after which they were centrifuged at 3,000 r/min by a laboratory physician (*Luo et al., 2022*). The liver function and blood lipid levels were measured using a fully automated biochemical analyzer (Cobas 701; Roche, Basel, Switzerland). Adverse pregnancy outcomes included caesarean section, preeclampsia (*Sinkey et al., 2020*) (SBP $\geq$ 140 mmHg and/or DBP $\geq$ 90 mmHg with urinary protein $\geq$0.3g/24 h or random urine protein), premature rupture of membranes (PROM) (*Ronzoni et al., 2022*), meconium-stained amniotic fluid (MSAF) (*Estiú et al., 2017*), postpartum hemorrhage (*Schlembach et al., 2014*) ($\geq$500 mL for vaginal deliveries and $\geq$1000 mL for deliveries by caesarean section), preterm (gestational age < 37 weeks), neonatal hypoglycemia (blood glucose < 2.2 mmol/L), neonatal respiratory distress (*Bricelj et al., 2017*), neonatal hyperbilirubinemia (*Maisels et al., 2009*), and cardiac injury.

### Diagnosis of ICP

ICP is diagnosed when the level of TBA in a pregnant woman increases by $\geq$ 10 $\mu$mol/L during pregnancy. Patients were classified as having mild or severe cholestasis based on TBA concentrations of 10–40 or $\geq$ 40 $\mu$mol/L, respectively (*Mays, 2010*; *Liu et al., 2015*).

### Statistical analysis

Data were collected as previously study described in *Fan et al. (2024)*. Specifically, statistical analysis of the data was carried out using SPSS 22.0 software. Continuous variables were

described by median (range), and frequencies (percentages) were described as categorical variables. The Mann–Whitney U test was used to analyze the differences between non-normal continuous variables and the chi-square test was used to compare the differences between categorical variables. Binary logistic regression analysis was used to evaluate the relationship between adverse pregnancy outcomes and ALT, AST, ALT/AST, and HDL respectively. The association between the levels of the above indicators and the incidence of adverse pregnancy outcomes were evaluated by odds of exposure (OR) and 95% confidence intervals (CI) to assess the precision of the point estimates. The receiver operating characteristic (ROC) curves for ALT/AST independently and combined with HDL were both drawn, and the sensitivity and specificity were calculated according to ROC curves. All statistical analyses with significant differences were considered to be $p < 0.05$.

# RESULTS

## Study population description

According to their serum total bile acid (TBA) levels, 147 pregnant women with ICP had TBA levels $\geq 10\ \mu mol/L$ were participate in the study, of which 122 mild ICP women had TBA levels of 10-39.9 $\mu mol/L$, and 25 severe ICP women had TBA levels $\geq 40\ \mu mol/L$. 120 normal pregnant women with TBA level <10 $\mu mol/L$ during the same period were selected as the control group. The general information of 267 included pregnant women were shown in Table 1. The characteristics of enrolled pregnant women such as age, pre-pregnancy BMI, reproductive history, and the gender of neonates had no significant differences. The neonatal birth weight in pregnant women with ICP was significantly lower than those in normal pregnant women ($p < 0.05$). We found the gestational week in the mild and severe ICP women were all significantly lower compared to the normal group ($p < 0.05$) and the gestational age at diagnosis in the severe ICP group was also lower than in the mild ICP group ($p < 0.05$). Although there was no significant difference in gestational weight gain (GWG) between ICP and normal pregnant women, a lower GWG was detected in the severe ICP women than in the mild ICP group ($p < 0.05$).

## Comparison of biochemical indices of mild ICP, severe ICP and normal pregnant women

We compared the laboratory biochemical indices among the mild ICP group, the severe ICP group and the normal group in Table 2. We found lower levels of total protein (TP) and albumin (ALB), and the blood urea nitrogen (BUN), serum creatinine (CR) and uric acid (UA) levels were higher in the ICP group compared to the normal group (all $p < 0.05$). We also detected the significantly higher levels of TBIL, DBIL and IBIL in the severe ICP group than in the mild ICP group (all $p < 0.05$). To our surprise, although the levels of TG, TC and LDL were not significantly different in the normal, mild ICP and severe ICP groups ($p > 0.05$), we discovered significant differences levels of AST, ALT, AST/ALT, A/G and HDL not only in the normal and ICP groups, but also in the different TBA levels of ICP groups. In particular, in Supplemental Information 1, increasing TBA level was associated with higher AST, ALT, ALT/AST and lower HDL level (all $p < 0.05$).

**Table 1  General information of participants.**

| Variables | Normal (TBA < 10 µmol/L) | Mild ICP (TBA 10–39.9 µmol/L) | Severe ICP (TBA ≥ 40 µmol/L) |
|---|---|---|---|
| n | 120 | 122 | 25 |
| Age (years) | 29.0 (27.0–32.0) | 32.0 (30.0–35.0) | 32.5 (31.0–34.0) |
| Pre pregnancy BMI (kg/m²) | 20.8 (19.5–22.9) | 24.5 (19.6–24.0) | 22.2 (20.4–22.7) |
| Gestational weight gain (kg) | 14.0 (12.0–17.0) | 15.0 (11.5–19.0)[b] | 12.0 (8.0–15.0)[b] |
| Nulliparous (n%) | 76.0 (63.3) | 80.0 (65.6) | 12 (48.0) |
| Gestational week (week) | 39.4 (38.7–40.1) | 38.3 (36.0–39.4)[a] | 36.1 (35.0–37.6)[a,b] |
| Gestational age at diagnosis (week) | – | 37.0 (34.0–38.0)[b] | 33.0 (30.0–36.0)[b] |
| Male neonates (n%) | 68 (56.7) | 65 (53.3) | 15 (60.0) |
| Female neonates (n%) | 52 (43.3) | 57 (46.7) | 10 (40.0) |
| Neonatal birth weight (grams) | 3300 (3050–3500) | 2905 (2387.5–3367.5)[a] | 2750 (2300–2950)[a] |

Notes.

BMI, body Mass Index; ICP, intrahepatic cholestasis of pregnancy; TBA, total bile acid.

All continuous variables in the table were given as the medians (quartile 1–quartile 3), except for women with nulliparous and gender of neonates values given n (%).

Mann–Whitney U test for the continuous variables and Chi-square test for the categorical variables were used to compare the general information between the pregnant women with mild and severe ICP and normal pregnant.

[a] $p < 0.05$, mild-ICP/sever-ICP compared with normal pregnancy.

[b] $p < 0.05$, sever-ICP compared with mild-ICP.

**Table 2  Laboratory biochemical indices of participants.**

| Variables | Normal (TBA <10 µmol/L) | Mild ICP (TBA 10–39.9 µmol/L) | Severe ICP (TBA ≥ 40 µmol/L) |
|---|---|---|---|
| ALT (U/L) | 9.0 (7.0–12.0) | 12.0 (9.0–32.0)[a] | 118.0 (58.75–427.5)[a,b] |
| AST(U/L) | 16.0 (14.0–19.0) | 22.0 (17.0–42.0)[a] | 108.0 (42.8–435.5)[a,b] |
| ALT/AST | 0.56 (0.47–0.69) | 0.59 (0.47–0.84)[a] | 0.94 (0.60–1.52)[a,b] |
| TBIL (µmol/L) | 7.8 (6.5–10.0) | 8.4 (5.9–10.1) | 17.9 (10.8–23.7)[b] |
| DBIL (µmol/L | 1.3 (1.0–1.6) | 1.5 (1.0–2.5) | 8.4 (2.5–12.0)[b] |
| IBIL (µmol/L) | 6.6 (5.5–8.3) | 6.4 (5.2–8.3) | 8.7 (6.2–11.0)[b] |
| TP (g/L) | 62.1 (60.2–64.8) | 60.9 (57.6–64.6)[a] | 61.1 (57.6–65.7)[a] |
| ALB (g/L) | 36.6 (35.4–37.8) | 35.1 (32.4–37.5)[a] | 35.3 (30.7–37.2)[a] |
| A/G | 1.4 (1.3–1.5) | 1.4 (1.2–1.4)[a] | 1.15 (1.1–1.4)[a,b] |
| BUN (mmol/L) | 2.7 (2.2–3.1) | 3.4 (2.8–4.7)[a] | 3.1 (2.4–4.6)[a] |
| CR (mmol/L) | 48.8 (43.9–55.6) | 55.0 (46.8–70.3)[a] | 50.2 (46.0–95.5)[a] |
| UA (mmol/L) | 244.0 (203.5–302.0) | 324.0(236.0–427.0)[a] | 290.0 (179.0–548.3)[a] |
| HDL (mmol/L) | 2.0 (1.7–2.2) | 1.8 (1.6–2.1)[a] | 1.2 (0.9–1.9)[a,b] |
| TG (mmol/L) | 3.0 (2.4–3.7) | 3.2 (2.6–4.4) | 3.0 (2.3–4.5) |
| TC (mmol/L) | 6.2 (5.4–7.0) | 6.6 (5.3–7.1) | 5.6 (4.3–6.6) |
| LDL (mmol/L) | 3.5 (3.9–4.1) | 3.8 (3.0–4.3) | 3.4 (2.6–4.2) |

Notes.

ALT, alanine aminotransferase; AST, aspartate aminotransferase; ALB, serum albumin; A/G, serum albumin/globulin; BUN, blood urea nitrogen; CR, creatinine; DBIL, direct bilirubin; HDL, high-density lipoprotein; IBIL, indirect bilirubin; LDL, low-density lipoprotein; TBIL, total bilirubin; TC, total cholesterol; TG, triglyceride; TP, serum total protein; UA, uric acid.

All continuous variables in the table were given as the medians (quartile 1–quartile 3) and we used Mann–Whitney U test to compare the differences between the pregnant women with mild/severe ICP and normal pregnant.

[a] $p < 0.05$, mild ICP/sever ICP compared with normal pregnancy.

[b] $p < 0.05$, sever ICP compared with mild ICP.

**Table 3  Pregnancy outcomes with mild and severe ICP and normal pregnant women.**

| | Adverse outcome (*n*%) | Normal (TBA < 10 μmol/L) | Mild ICP (TBA 10–39.9 μmol/L) | Severe ICP (TBA ≥ 40 μmol/L) |
|---|---|---|---|---|
| Maternal | Cesarean section | 41 (34.2) | 75 (61.5)[a] | 23 (92.0)[a,b] |
| | Preeclampsia | 2 (1.7) | 39 (32.0)[a] | 2 (8.0)[a,b] |
| | PROM | 29 (24.2) | 14 (11.5)[a] | 1 (4.0)[a] |
| | MSAF | 1 (0.8) | 19 (15.6)[a] | 9 (36.0)[a,b] |
| | Postpartum hemorrhage | 0 (0.0) | 2 (1.6) | 0 (0.0) |
| Neonatal | Premature birth | 4 (3.3) | 31 (25.4)[a] | 13 (52.0)[a,b] |
| | Hypoglycemia | 1 (0.8) | 4 (3.3) | 1 (4.0)[b] |
| | Respiratory distress | 0 (0.0) | 8 (6.6)[a] | 2 (8.0)[a] |
| | Hyperbilirubinemia | 11 (9.2) | 32 (26.2)[a] | 13 (52.0)[a,b] |
| | Cardiac injury | 10 (8.3) | 21 (17.2)[a] | 9 (36.0)[a,b] |

Notes.

PROM, premature rupture of membranes; MSAF, meconium-stained amniotic fluid.

Chi-square test and Fisher's Exact test for the categorical variables were used to compare the adverse maternal and infant pregnancy outcomes in the mild ICP, severe ICP and normal pregnant groups.

[a]$p < 0.05$, mild-ICP/sever-ICP compared with normal pregnancy.

[b]$p < 0.05$, sever-ICP compared with mild-ICP.

## Comparison of pregnancy outcomes between mild and severe ICP and normal pregnant women

As shown in Table 3, except for postpartum hemorrhage, the rate of adverse maternal pregnancy outcomes such as cesarean section, preeclampsia, PROM, and MSAF in the ICP group were all significantly different from those in the normal group (all $p < 0.05$), and the significant differences of cesarean section, preeclampsia, and MSAF were also found between the mild ICP group and severe ICP group (all $p < 0.05$). Comparison of neonatal outcomes between normal and ICP groups revealed significant differences in neonatal preterm, hypoglycemia, respiratory distress, hyperbilirubinemia, and cardiac injury (all $p < 0.05$). Surprisingly, as maternal TBA levels increased, there were obvious higher rate of neonatal preterm [31(25.4%) *vs.* 13 (52.0%), ($p = 0.008$)], hyperbilirubinemia [32 (26.2%) *vs.* 13 (52.0%), ($p = 0.011$)] and cardiac injury [21 (17.2%) *vs.* 9 (36.0%) ($p = 0.034$)] in the severe ICP group than in the mild ICP group.

## Associations between biochemical indices and adverse outcomes of maternal and neonatal

ICP causes liver function injury and lipid metabolism disorders in pregnant women, which not only increases the incidence of adverse maternal outcomes (*Menzyk et al., 2018*; *Zhan et al., 2022*), but also has a serious impacts on fetal development (*Estiú et al., 2017*; *Pillarisetty & Sharma, 2023*; *Koh, Kathirvel & Mathur, 2021*). A similar result was presented in our study, the ICP group had a significantly more severe liver function impairment, lower levels of HDL (Table 2 and Supplemental Information 1), and a higher incidence of adverse maternal and neonatal outcomes (Table 3). Then, we further performed a logistic regression analysis of the associations between ALT, AST, ALT/AST, HDL and the risk of adverse maternal and neonatal outcomes (Supplemental Information 2 and Table 4). The results of adverse maternal outcomes showed that ALT, AST, and ALT/AST were associated

**Table 4  Associations between biochemical indices and adverse neonatal outcomes.**

| Adverse outcome | Variables | Model 1 | | Model 2 | |
|---|---|---|---|---|---|
| | | OR (95% CI) | *p*-value | AOR (95% CI) | *p*-value |
| Premature birth | ALT | 1.005 (1.002–1.008) | 0.003 | 1.003 (1.000–1.006) | 0.044 |
| | AST | 1.004 (1.001–1.007) | 0.017 | 1.002 (0.999–1.006) | 0.160 |
| | ALT/AST | 7.575 (3.179–18.050) | 0.000 | 4.696 (1.859–11.861) | 0.001 |
| | HDL | 0.151 (0.053–0.428) | 0.000 | 0.277 (0.093–0.826) | 0.051 |
| Hypoglycemia | ALT | 1.000 (0.992–1.008) | 0.961 | 0.998 (0.987–1.009) | 0.759 |
| | AST | 1.000 (0.991–1.009) | 0.983 | 0.997 (0.985–1.010) | 0.674 |
| | ALT/AST | 3.303 (1.042–10.472) | 0.042 | 3.554 (0.931–13.574) | 0.064 |
| | HDL | 0.149 (0.014–1.637) | 0.120 | 0.430 (0.034–5.470) | 0.515 |
| Respiratory distress | ALT | 1.003 (0.999–1.007) | 0.096. | 1.002 (0.998–1.006) | 0.363 |
| | AST | 1.001 (0.996–1.006) | 0.669 | 0.999 (0.992–1.006) | 0.840 |
| | ALT/AST | 6.013 (2.229–16.226) | 0.000 | 4.366 (1.452–13.126) | 0.009 |
| | HDL | 0.127 (0.028–0.577) | 0.008 | 0.222 (0.047–1.046) | 0.057 |
| Hyperbilirubinemia | ALT | 1.004 (1.001–1.006) | 0.016 | 1.002 (1.000–1.005) | 0.095 |
| | AST | 1.002 (1.000–1.005) | 0.093 | 1.001 (0.999–1.004) | 0.335 |
| | ALT/AST | 5.184 (2.338–11.494) | 0.000 | 4.019 (1.757–9.194) | 0.001 |
| | HDL | 0.226 (0.092–0.554) | 0.001 | 0.315 (0.126–0.788) | 0.014 |
| Cardiac injury | ALT | 1.004 (1.001–1.007) | 0.006 | 1.003 (1.000–1.006) | 0.029 |
| | AST | 1.002 (0.999–1.005) | 0.158 | 1.001 (0.998–1.004) | 0.395 |
| | ALT/AST | 4.516 (2.056–9.920) | 0.000 | 3.500 (1.535–7.987) | 0.003 |
| | HDL | 0.143 (0.048–0.427) | 0.000 | 0.134 (0.039–0.461) | 0.001 |

**Notes.**

OR, odds ratio; AOR, adjusted odds ratio.

Logistic regression analysis was performed to determine the associations between biochemical indices and adverse neonatal outcomes. Model 1 was unadjusted for confounding factors; Model 2 was adjusted for maternal gestational age (<30 reference, >30), pre pregnancy BMI (<24 reference, >24), gestational weight gain, reproductive history and gestational age at diagnosis.

with the risk of adverse maternal outcomes of cesarean section and MSAF in either Model 1 (univariate regression analysis) and Model 2 (adjusted for maternal gestational age, pre-pregnancy BMI, GWG, reproductive history, and gestational age at diagnosis) (all $p < 0.05$), we found the same association for HDL in Model 1 ($p < 0.05$), but no significant association in Model 2, which adjustment for confounding (all $p > 0.05$) (Supplemental Information 2).

Additionally, we also performed a logistic regression analysis of the correlation between ALT, AST, ALT/AST, HDL and the risk of adverse neonatal outcomes (Table 4). We surprisingly found that ALT/AST was a significant risk factor for the development of premature birth [adjusted odds ratio (AOR) = 4.696, 95% CI [1.859–11.861], $p = 0.001$], neonatal respiratory distress [adjusted odds ratio (AOR) = 4.366, 95% CI [1.452–13.126], $p = 0.009$], neonatal hyperbilirubinemia [adjusted odds ratio (AOR) = 4.019, 95% CI [1.757–9.194], $p = 0.001$] and cardiac injury [AOR = 3.500, 95% CI [1.535–7.987], $p = 0.003$] respectively in both univariate and multiple regression analyses. Meanwhile, as the protective factor for the development of neonatal hyperbilirubinemia and cardiac injury,

HDL was also found in regression analyses [AOR = 0.315, 95% CI [0.126–0.788], $p$ = 0.014; AOR = 0.134 (0.039–0.461), $p$ = 0.001].

**Exploring the predictive value of serum ALT/AST combined with HDL for adverse neonatal outcomes in ICP pregnant women**

Table 2 showed that ALT/AST and HDL levels were significantly different between the normal group and the different degrees of ICP groups, and both were associated with the development of neonatal hyperbilirubinemia and cardiac injury (Table 4). To assess the combined predictive efficacy of ALT/AST and HDL levels on adverse neonatal outcomes, the ROC analysis were plotted in the Fig. 1, which showed the AUC value for ALT/AST independently predicted adverse neonatal hyperbilirubinemia outcome was 0.644 [95% confidence interval (CI): 53.6–75.2%, $p$ = 0.008], the sensitivity and specificity were 50.0% and 77.3% respectively. The combined predictive efficacy of ALT/AST and HDL levels increased the AUC (0.668) [95% CI [56.3–77.3]%, $p$ = 0.002] and the sensitivity and specificity were 47.1% and 84.0%, respectively (Fig. 1A). Similarly, we described ROC curve for ALT/AST and HDL for predicting adverse outcomes in neonatal cardiac injury, Fig. 1B showed that the combination of HDL levels predicted neonatal cardiac injury with the AUC of 0.668 [95% CI [56.4–77.1]%, $p$ = 0.002], and the sensitivity and specificity were 41.2% and 87.1%, respectively, all of which were more efficient than ALT/AST alone in predicting neonatal cardiac injury (AUC: 0.644 (95% CI [53.6–75.2]%, $p$ = 0.008), sensitivity and specificity were 50.0% and 77.3% respectively). In order to further clarify the predictive value of ALT/AST and HDL, we analyzed the value of TBA in predicting neonatal hyperbilirubinemia and cardiac injury in Supplemental Information 3, the results showed that the value of TBA in predicting neonatal hyperbilirubinemia and cardiac injury was lower than that of ALT/AST and HDL (AUC: 0.653, 95% CI [54.5–76.0]%, $p$ = 0.005, and AUC: 0.662, 95% CI [55.3–77.2]%, $p$ = 0.003). These findings suggested that ALT/AST combined with HDL may be a predictive clinical value of adverse neonatal outcomes of hyperbilirubinemia and cardiac injury.

## DISCUSSION AND CONCLUSIONS

With the increased incidence of ICP, early diagnosis of the diseases during pregnancy is increasingly important. Most studies support the use of serum TBA as a primary indicator for the diagnosis of ICP in clinical work due to its toxic effects at high concentrations, which can lead to ICP-related adverse pregnancy outcomes (Wood et al., 2018; Smith & Rood, 2020; Bicocca, Sperling & Chauhan, 2018). Previous studies have shown that TBA has some cytotoxic effects, causing liver cell apoptosis and necrosis, and leading to disorders of liver function, such as increased levels of ALT, AST, and TBIL (Zhu et al., 2019). Our study has consistently concluded that liver function indicators showed varying degrees of abnormality in the ICP group, with significantly higher levels of TBIL in the severe ICP group. In particular, the levels of ATL, AST, and ALT/AST were significantly different between the normal group and the ICP group. By further correlation analysis of Spearman,

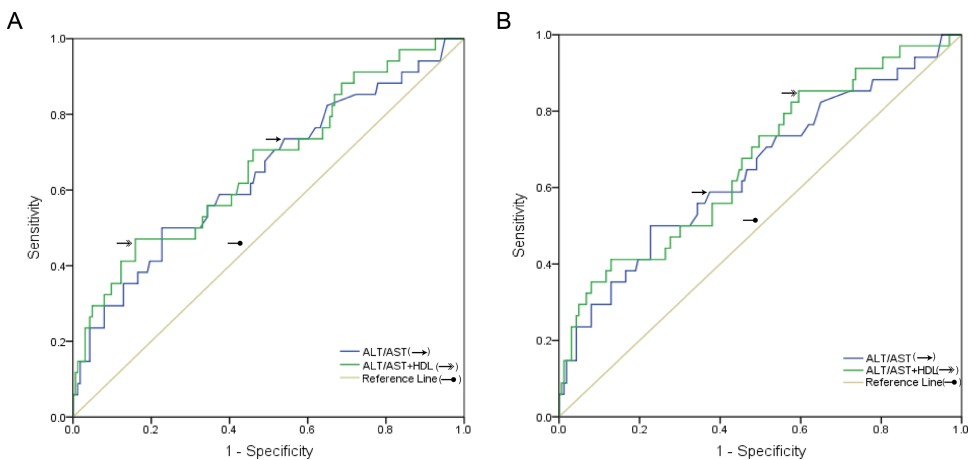

**Figure 1  Sensitivity and specificity of ALT/AST combined with HDL in the assessment of ROC curves to predict adverse pregnancy outcomes in neonatal hyperbilirubinemia and cardiac injury.**

we found that with the increase of TBA level, these indicators also increased significantly, which was positively correlated with the severity of ICP.

*Marschall et al. (2013)*, through a large cohort study, found that ICP patients had a significantly increased risk of hepatobiliary disease after pregnancy. *Jelski et al. (2020)* further found that there were significant changes in some markers of liver cell enzyme activity in the serum of pregnant women with ICP, which could be used as potential markers to evaluate ICP (*Jelski et al., 2020*; *Piechota et al., 2021*; *Piechota et al., 2022*). For example, in addition to the significant increase of the TBA, the alcohol dehydrogenase isoenzyme (ADH I) activity was also significantly increased in ICP patients. Further, it was concluded that the sensitivity and specificity of ADH I in the diagnosis of ICP were as high as 85% and 91%, respectively, which is of great value in the diagnosis of ICP. Therefore, although our study did not focus on hepatocyte enzyme activity, it suggests that we explore changes in metabolic markers related to liver function in pregnant women with ICP, such as those related to lipid metabolism, which may lead to more novel and effective markers for the diagnosis of ICP.

According to the literatures, ICP is related to dyslipidemia in pregnant women, which may be due to the high estrogen levels (*Xiao et al., 2021*). Abnormally elevated estrogen affects the normal metabolism of liver cells, leading to liver function injury, and aggravates the abnormal lipid metabolism in pregnant women while inducing cholestasis. However, the specific role of lipid metabolism in the pathogenesis of ICP is unclear, and the correlation between lipid metabolism and ICP is rarely reported. Therefore, we evaluated the diagnostic value of lipid metabolism indicators in pregnant women with different ICP levels and normal pregnant women. Although there were no statistical differences in TG, TC, and LDL, the levels of lipid metabolism indicators were elevated in the ICP group. We also noted that HDL, as a protective factor against cardiovascular, was significantly lower in the ICP group than in the control group, and further correlation analysis showed a negative correlation between HDL and TBA levels. Our study supports the conclusion that serum

lipid metabolism is disturbed in ICP patients. Unfortunately, due to the lack of data on estrogen and progesterone during pregnancy, it is impossible to conduct an insightful analysis of the relationship between liver function, estrogen and progesterone, and lipid metabolism and ICP.

As a special complication of pregnancy, ICP can trigger adverse effects on both the pregnant women and the fetus. Previous studies have reported, pregnant women with ICP have an increased risk of cesarean section, preeclampsia, and postpartum hemorrhage due to the toxic effects of TBA (*Menzyk et al., 2018*; *Zhan et al., 2022*; *Celik et al., 2019*). Our study also found higher rates of cesarean section and preeclampsia in ICP patients than in women with normal pregnancies, but no difference in postpartum hemorrhage. Although ICP is a relatively benign disease for the mother, it is strongly associated with adverse perinatal outcomes, including premature birth, respiratory distress, hyperbilirubinemia, and other adverse outcomes (*Estiú et al., 2017*; *Pillarisetty & Sharma, 2023*; *Koh, Kathirvel & Mathur, 2021*). In previous studies, a positive association was found between adverse neonatal outcomes and maternal serum TBA levels in patients with ICP (*Kawakita et al., 2015*; *Liao et al., 2023*). Cohort studies have confirmed that increased maternal serum TBA concentration is positively correlated with the incidence of premature birth, possibly due to the maternal placental villi exhibit a dependent contraction response in the presence of elevated bile acid levels, which affects the fetal-maternal nutrient exchange, resulting in ischemia and hypoxic symptoms and inducing neonatal respiratory distress and other syndromes (*Mascio et al., 2021*; *Wang et al., 2019*).

In addition, impaired maternal bile excretion and absorption may lead to chronically elevated bilirubin and bile acid levels in the fetus, causing neonatal neurological damage and increasing the risk of adverse pregnancy outcomes such as cerebral palsy and intellectual impairment (*Huang et al., 2021*; *Qian, Kumar & Testai, 2022*; *Shapiro, 2003*). In our findings, the incidence of premature birth, neonatal respiratory distress, and hyperbilirubinemia in the ICP group were higher than in women with normal pregnancies, and in particular, the incidence of cardiac injury in newborns of ICP women was found to increase significantly with higher maternal serum TBA levels. The influence of maternal ICP on the neonatal cardiovascular system has attracted more attention in recent years. *Song, Tian & Shi (2021)*, *Williamson et al. (2001)*, *Ovadia & Williamson (2016)* and *Kowalski & Abelmann (1953)* showed that the increase in fetal intrauterine death in ICP patients is related to fetal cardiac dysfunction and arrhythmia induced by bile acid accumulation. Our study also demonstrated a higher rate of neonatal cardiac injury and greater adverse effects on the neonatal cardiovascular system with increasing maternal TBA concentrations. We acknowledge that there may be errors in our study due to the limitations of retrospective studies, as active management of cholestasis, such as treatment of ursodeoxycholic acid (UDCA), may bias the results of these studies and not fully reflect the truth of the population, but our data still provide some clinical basis for understanding the mechanism of ICP-related neonatal heart disease and confirm the hypothesis that elevated maternal bile acid induced neonatal cardiac injury (*Liao et al., 2023*).

We know that ICP related adverse perinatal outcomes have a significant adverse impact on newborns. However, there is no consensus on the diagnostic threshold of biochemical

indicators such as liver enzymes, which may limit the early diagnosis of ICP and prevents early intervention to reduce the occurrence of adverse neonatal pregnancy outcomes. Therefore, we used ROC curves to predict and evaluate the diagnostic value of the liver enzymes indicator ALT/AST and the lipid metabolism indicator HDL for adverse perinatal outcomes in pregnant women with ICP. The ALT/AST had certain predictive values for both neonatal hyperbilirubinemia and adverse outcomes of cardiac injury, however, the combined detection of the ALT/AST and HDL level improved the predictive efficacy and specificity. To summarize, combined ALT/AST and HDL have excellent diagnostic value for ICP related adverse perinatal outcomes and further, provide evidence for the feasibility of timely intervention to reduce adverse neonatal pregnancy outcomes. However, the main flaw of this study was the absence of comprehensive data on medication use during pregnancy and the lack of stratification of lipid metabolism indicators according to factors such as daily dietary habits. In the end, we need further large sample size retrospective studies or large cohort prospective studies to validate our findings so that they have clinical use and merit clinical development.

## ACKNOWLEDGEMENTS

We thank all subjects and participants for their support and cooperation in this study and for their cooperation and participation in the study.

### Funding

The study was funded by the Key Medical Scientific Research Project of Hebei province (No. 20231652) and the Hebei Provincial Department of Human Resources and Social Security (No. C20200358). The funders had no role in study design, data collection and analysis, decision to publish, or preparation of the manuscript.

### Grant Disclosures

The following grant information was disclosed by the authors:
Key Medical Scientific Research Project of Hebei province: No. 20231652.
Hebei Provincial Department of Human Resources and Social Security: No. C20200358.

### Competing Interests

The authors declare there are no competing interests.

### Author Contributions

- Xizhenzi Fan conceived and designed the experiments, performed the experiments, analyzed the data, prepared figures and/or tables, authored or reviewed drafts of the article, and approved the final draft.
- Xia Li conceived and designed the experiments, performed the experiments, analyzed the data, prepared figures and/or tables, authored or reviewed drafts of the article, and approved the final draft.

- Tianxiao Yu conceived and designed the experiments, prepared figures and/or tables, authored or reviewed drafts of the article, and approved the final draft.
- Ruifen Jiao performed the experiments, authored or reviewed drafts of the article, administrative technical or material support, and approved the final draft.
- Wenhui Song analyzed the data, prepared figures and/or tables, and approved the final draft.
- Achou Su performed the experiments, prepared figures and/or tables, and approved the final draft.
- Mingwei Li performed the experiments, prepared figures and/or tables, and approved the final draft.
- Qing Guo conceived and designed the experiments, authored or reviewed drafts of the article, supervision, and approved the final draft.

## Human Ethics

The following information was supplied relating to ethical approvals (*i.e.*, approving body and any reference numbers):

This study was approved by the ethics committee of The Fourth Hospital of Shijiazhuang (No 20230034).

## Data Availability

The raw data is available in the Supplemental File.

## Supplemental Information

Supplemental information for this article can be found online at http://dx.doi.org/10.7717/peerj.17613#supplemental-information.

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
