# Peer review of "Evaluation of alanine aminotransferase/aspartate aminotransferase ratio and high-density lipoprotein for predicting neonatal adverse outcomes associated with intrahepatic cholestasis of pregnancy"

_PeerJ, doi:10.7717/peerj.17613_

## Round 0.1 · original submission · Major Revisions

Please address issues indicated by the reviewers and amend manuscript accordingly.

·

Basic reporting

The study is based on extensive and recent literature, gives some new information and this warrants its publication.

Experimental design

The study is well done, the material is large enough and the methods look reliable.

Validity of the findings

This well written manuscript addresses an important topic exploring diagnostics of intrahepatic cholestasis of pregnancy.

Additional comments

Although several reviews about the diagnostics of ICP have been already published, the discussion on the markers of ICP in this paper seems to be original. However I have the following suggestions/comments and hope the authors can address them in the review.
Minor revision
Some authors showed that the ADH/ALDH activities are higher in of women with I|CP, suggesting that isoenzymes of ADH have a valuable use as a diagnostic marker for ICP
1. Jelski Wojciech, Piechota Joanna, Orywal Karolina, Mroczko Barbara: The alterations in of alcohol dehydrogenase activity in the sera of women with intrahepatic cholestasis of pregnancy. Anticancer Res. 2020, 40, 1997-2001.
2. Piechota Joanna, Jelski Wojciech, Orywal Karolina, Mroczko Barbara. The comparison of total bile acid concentration and alcohol dehydrogenase activity as markers of intrahepatic cholestasis of pregnancy. Acta Biochimica Polonica. 2022; 69:173-176
3. Piechota Joanna, Jelski Wojciech, Orywal Karolina, Mroczko Barbara. The alcohol dehydrogenase isoenzyme (ADH I) as a marker of intrahepatic cholestasis of pregnancy. Scientific Reports 2002: 12: 11071

Please discuss (5-6 sentences)

·

Basic reporting

The wording and logical flow is quite clear and easy to follow.
Here are a few detailed points to improve:
1) Introduction: Please provide basic background on the ALT/AST biochemical function and normal levels in healthy humans.
2) Line 157: TP,ALB, BUN, CR, and UA are mentioned first; maybe include their full names.
3) Line 180 to 189 is a long discussion; maybe it would be better to put it in the discussion section.

Experimental design

The work is solid. The sample size, statistics, and modeling are valid experimental designs.
Is it possible to compare the ALT/AST and HDL level to patents with liver disease, to see if liver disease and ICP can be distinguished.

Validity of the findings

The authors described a new diagnostic approach to predicting the risk of ICP. This ALT/AST and HDL level-based approach is novel, and hence, it has an impact that this journal expected.

Additional comments

4) As ALT/AST and HDL are all indicators for liver disease, have you studied how to distinguish other liver disease and ICP.

Reviewer 3 ·

Basic reporting

This study showed a correlation of between the level of maternal alanine aminotransferase/aspartate aminotransferase (ALT/AST) and high-density lipoprotein (HDL) and the disease severity of ICP in pregnant women by analyzing the level of ALT/AST and HDL in a patient population of more than a hundred people. They also hypothesized that there could be potential value in using the expression level of ALT/HDL and HDL as the predictors of adverse neonatal outcomes of pregnant women with ICP. Overall, the paper did a good job of acquiring the data in the patient population and analyzing the data.

Experimental design

Overall, I consider this manuscript to be technically sound in the experiments and analysis. However, the novelty and workload do not meet the standard that the PeerJ is looking for

Validity of the findings

I consider that this paper does not meet the quality standard of PeerJ for below reasons:

1. This paper lacks novelty of presenting anything significantly new that has not been reported. As the author mentioned, the level of ALT/AST and HDL has been validated as indicators for ICP-related adverse pregnancy outcomes by many reports before. The analysis provided here in this paper only provided incremental new discoveries of more detailed association between the expression level and the adverse neonatal outcomes. Meanwhile, the author pointed out that a positive correlation between the level of TBA and adverse neonatal outcomes but they did not show whether using ALT/AST and HDL could be better predicator than the TBA level. This level of novelty does not match the paper quality that PeerJ is looking for.
2. The paper feels incomplete and the workload does not meet the standard of the PeerJ. The author argued that the expression level of ALT/AST and HDL may serve as the predictors of adverse neonatal outcomes. However, there is no evidence of showing the use of the expression level of the three as the diagnostic tool. One experiment can be done is to set up a standard based on the expression level of ALT/AST and HDL and classify the patient based on the standard and compare it to the classical diagnosis based on the level of TBA. This will help to determine the diagnostic value of the expression level of ALT/AST and HDL.
3. Other minor issues include the big difference between the sample size of the severe ICP and normal patients and there are some sentences that are not well-written in the manuscript.

---

## Round 0.2 · accepted · Accept

Issues pointed by the reviewers were adequately addressed and the revised manuscript is acceptable now.

·

Basic reporting

The authors clearly addressed my review comments last time. They made modifications to the text and also provided excellent explanations for some of my concerns and questions.

Experimental design

The authors clearly explained why they did not compare the ALT/AST and HDL level of pregnant and liver disease patients. I think their explanation is clear and reasonable.

Validity of the findings

This newly developed diagnostic method is a novel finding, and has good applications to ICP early diagnostic.

Additional comments

They did a sufficient revision to their manuscript, and I would recommend an acceptance.